# Investigating Factors Associated with Fear of Falling in Community-Dwelling Older Adults through Structural Equation Modeling Analysis: A Cross-Sectional Study

**DOI:** 10.3390/jcm12020545

**Published:** 2023-01-09

**Authors:** Elane Priscila Rosa dos Santos, Daniela Gonçalves Ohara, Lislei Jorge Patrizzi, Isabel Aparecida Porcatti de Walsh, Caroline de Fátima Ribeiro Silva, José Ribeiro da Silva Neto, Nayara Gomes Nunes Oliveira, Areolino Pena Matos, Natalia Camargo Rodrigues Iosimuta, Ana Carolina Pereira Nunes Pinto, Maycon Sousa Pegorari

**Affiliations:** 1Department of Biological and Health Sciences, Federal University of Amapá, Road Juscelino Kubitschek, Km-02, Jardim Marco Zero, Macapá 68903-419, Brazil; 2Department of Applied Physiotherapy, Federal University of Triângulo Mineiro, Boulevard Frei Paulino, no 30, Abadia, Uberaba 38025-180, Brazil; 3Postgraduate Program in Physical Therapy, Federal University of Triângulo Mineiro, Boulevard Frei Paulino, no30, Abadia, Uberaba 38025-180, Brazil; 4Postgraduate Program in Health Sciences, Federal University of Amapá, Road Juscelino Kubitschek, Km–02, Jardim Marco Zero, Macapá 68903-419, Brazil; 5Department of Nursing in Education and Community Health, Federal University of Triângulo Mineiro, Boulevard Frei Paulino, no 30, Abadia, Uberaba 38025-180, Brazil

**Keywords:** aged, health of the elderly, fear of falling, accidental falls, latent class analysis, models, statistical

## Abstract

The scientific literature mentions the existence of factors associated with fear of falling in older adults. However, the direct and indirect paths of its predictors have not yet been fully explored. This study aimed to analyze the socioeconomic, clinical, and health factors directly and indirectly associated with fear of falling in community-dwelling older adults. This is a cross-sectional study conducted in older adults (*n* = 410 – 70.11 ± 7.22 years). Clinical and health-condition data were collected, as were data on fear of falling using the Falls Efficacy Scale International—Brazil (FES-I Brazil). It was found that being female and having a higher number of self-reported morbidities, worse physical performance, and a higher number of depressive symptoms were directly associated with greater fear of falling. Regarding indirect associations, physical inactivity, mediated by a greater number of morbidities, worse physical performance, and a greater number of depressive symptoms, was associated with greater fear of falling. In addition, worse self-rated health, mediated by a greater number of depressive symptoms, as well as older age, mediated by worse physical performance, were associated with the outcome. This study provides information on the predictors directly and indirectly associated with fear of falling, expanding current understanding of this relationship.

## 1. Introduction

Falls and fall-related injuries are major public health concerns in older adults worldwide, as they represent an important burden to the person, their family, society, and the economy [1,2,3]. The continuous concern of an individual when standing or walking is associated with their degree of confidence when performing daily activities without falling [4]. Fear of falling is a psychological condition developed by older adults with and/or without a history of falls that may act not only as a cause, but also as a consequence of episodes of falls [5,6]. Fear of falling is associated with negative health outcomes in older adults, such as an increased risk of falls and trauma, which promotes limitations in daily physical activities, and deficits in physical performance, physical function, and mobility [5,6]. In this sense, it is suggested that the risk of falls reinforces concern about falls and may be related to a cascade of events that may ultimately result in social isolation, and an increased risk of hospitalization, and death in this population [7,8,9].

Some aspects of fall-related self-efficacy are based on Bandura’s social learning theory [10] and have been used to build scales to represent fear of falling during daily activities, such as the Falls Efficacy Scale—FES-I [4]. Under this perspective, performance mechanisms have shown a strong relationship with psychological changes and, consequently, with behavioral modifications, suggesting that self-efficacy acts on the behavioral decisions of dispositions [10]. Individuals who feel fear are more prone to avoiding circumstances that they see as difficult to face. On the other hand, when the same individuals see themselves as safe and capable, they are motivated to deal with intimidating situations and demonstrate confidence and engagement in activities. 

The prevalence of fear of falling varies according to each region, corresponding to 29% in the United States, 41.5% in Spain, 57.9% in Japan, 76.6% in Korea, and 78.2% in Portugal [6,9,11,12,13,14,15,16]. Among Brazilian studies, only studies conducted in the southern and southeastern regions have been performed, and the prevalence of fear of falling varies from 24 to 95%, according to data from cross-sectional studies [7,12,16,17,18,19,20,21]. The large variability among the prevalence of fear of falling across studies may be associated with the sample sizes used, the little-explored regions, the use of different instruments to assess concern about falling, and the diversity of the variables analyzed in association with fear of falling [8,9,12,16,19].

Previous studies have suggested that fear of falling is associated with factors such as history of falls, risk of falls, the use of more than two medications, activity restriction, dependence on daily activities, quality of life, loss of vision, self-assessment of health as fair or poor, and depressive symptoms [7,8,11,16,19,22]. Despite this, to our knowledge, no studies have explored the direct and indirect relationships of factors associated with fear of falling using more robust methods, such as the analysis of structural equation modeling.

In addition, studies on the prevalence of and factors associated with fear of falling in community-dwelling older adults in Brazil are concentrated in more developed regions with a greater population contingent. Fear of falling is a reversible condition [23,24,25]; therefore, studies that investigate the direct and indirect associations with socioeconomic, clinical, and health factors may aid in developing current understanding of the condition and the development of preventative and treatment strategies for this population. Therefore, the aim of this study was to analyze socioeconomic, clinical, and health factors directly and indirectly associated with fear of falling in community-dwelling older adults.

## 2. Materials and Methods

### 2.1. Study Context and Population

This was a cross-sectional study conducted in community-dwelling older adults living in the urban area of Macapá. The city of Macapá is bathed by the Amazon River, and is the capital of the only Brazilian state to be cut by the imaginary line of the equator, Amapá, located in the North region of Brazil [26]. With a life expectancy of 74.2 and a Human Development Index (HDI) of 0.733, in 2010, Macapá had an estimated population of 456,171 inhabitants. Of these, 5.21% (96% urban and 4% rural) were aged 60 years or older [27].

The sample population was based on estimated health problems in 50% of the older adult population, a 5% confidence interval, and a 95% confidence interval of a finite population of 19.955% of older adults, reaching a minimum required sample of 377 participants. A two-stage cluster sampling process was used to define the sample of this study. For this purpose, we considered the census sectors with information from the Brazilian Institute of Geography and Statistics (IBGE), regarding the neighborhoods and streets. With this information, we then identified the older adults in the residences [26]. This study was approved by the Human Research Ethics Committee of the Federal University of Amapá (CAAE: 57038316.9.0000.0003, Opinion 1,738,671). 

We included participants aged 60 years or older, of both sexes, living in the urban area of Macapá, and who were able to walk with or without a walking aid device. Older adults who were hospitalized and/or institutionalized with neurological diseases that prevented performance of the evaluations, and/or those who presented cognitive decline, were excluded. Cognitive decline was assessed through the Mini Mental State Examination (MMSE), using a version translated and validated in Brazil, considering the cut-off points related to education level [28]. Thus, of the 443 older adults evaluated, 27 were excluded due to cognitive decline and 6 for reasons such as insufficient information provided. Finally, this study was carried out in 410 community-dwelling older adults.

### 2.2. Fear of Falling (Dependent Variable)

We evaluated fear of falling using the Falls Efficacy Scale International—Brazil (FES-I Brazil) [4]. This scale is composed of 16 items pertaining to daily activities, related to concerns about falling, including domestic activities, personal care, leisure, and walking outside the home, and each item is scored from one to four. The total score is calculated with the sum of the scores obtained in each item, and thus, varies from 16 to 64, where lower values represent the absence of concerns about falling and higher scores represent greater concerns about falling [4].

### 2.3. Exploratory Variables

The following variables were considered: socioeconomic: age (in years) and sex (male and female); clinical and health: number of medications and morbidities, health perception (positive and negative), number of falls in the previous 12 months, and the use of a walking aid device (yes or no); depressive symptoms, assessed using the Abbreviated Geriatric Depression Scale (GDS-15) [29]; functional disability, assessed using the Katz Scale [30] for basic activities of daily living (BADL), and the Lawton and Brody Scale for instrumental activities of daily living (IADL) [20]; physical activity level, assessed using the International Physical Activity Questionnaire (IPAQ), adapted for older adults from Brazil [31,32]; and physical performance, assessed using the Short Physical Performance Battery (SPPB) [26,33]. 

### 2.4. Statistical Analysis

We used the Statistical Package for Social Sciences (SPSS^®^), version 24 and Analysis of Moment Structures (AMOS^®^), version 24 to analyze the data. The descriptive analysis considered the frequency distribution (absolute and relative) for categorical variables and measures of central tendency and dispersion (mean and standard deviation) for quantitative variables. To structure the model, it was considered that sociodemographic, clinical, and health characteristics were associated with greater fear of falling through direct and indirect trajectories. Thus, we elaborated a hypothetical model (Figure 1), composed of observed variables, represented by rectangles, and classified as endogenous and exogenous. We then tested this model through the analysis of trajectories [34]. For the endogenous variables, directional arrows are used and measurement errors have been attributed, denoted as “e” in the models [34].

We then carried out the steps for the analysis of the structural equation model: data collection, model estimation, and evaluation of the goodness of fit [34]. To estimate the parameters, we used the asymptotically distribution-free method and the goodness of fit of the model, according to: a Goodness of Fit Index (GFI) ≥ 0.95; a Tucker-Lewis Index (TLI) ≥ 0.90; a Chi-square test (χ^2^) *p* > 0.05; a Tucker-Lewis Index (TLI) ≥ 0.90; and a Root Mean Error of Approximation (RMSEA) ≤ 0.05 [34]. We tested the hypothetical model with subsequent specifications, to eliminate non-significant pathways (*p* > 0.05), and calculations of modification indices (≥11) [34]. We presented direct associations using estimates of standardized coefficients in the trajectories between sociodemographic, economic, and health variables and the number of depressive symptoms. We determined indirect associations using intermediate trajectories between these variables. In all tests, the type I error was fixed at 5% (*p* value <0.05).

## 3. Results

The prevalence of fear of falling among the evaluated community-dwelling older adults was 44%. The majority were female (66.3%), had fair self-rated health (55.4%), did not use a walking aid device (95.9%), and were insufficiently active (53.2%) (Table 1).

The means and standard deviations of the quantitative variables included in the model are described in Table 2.

Figure 2 presents a model of the association of socioeconomic, clinical, and health characteristics with fear of falling among community-dwelling older adults.

The direct estimators of the associations between the tested variables and the fear of falling among the community-dwelling older adults are shown in Table 3. It was found that female sex, a higher number of self-reported morbidities, worse physical performance assessed using the SPPB, and a higher number of depressive symptoms were directly associated with greater fear of falling.

Regarding indirect associations, it was identified that physical inactivity (β = 0.04; β = 0.02; β = 0.02), mediated by a higher number of morbidities, worse physical performance, and a higher number of depressive symptoms, was associated with greater fear of falling (Figure 2). Furthermore, worse self-rated health (β = −0.08), mediated by a greater number of depressive symptoms, as well as older age (β = 0.03), mediated by worse physical performance, were associated with the outcome (Figure 2).

## 4. Discussion

To our knowledge, this is the first study to demonstrate factors directly and indirectly associated with fear of falling in community-dwelling older adults using a structural equation modeling approach. In the present study, female sex, worse physical performance, a higher number of self-reported morbidities, and a higher number of depressive symptoms were directly associated with greater fear of falling. Additionally, older age, mediated by worse physical performance, as well as worse self-rated health, mediated by a higher number of depressive symptoms were associated with greater fear of falling.

The direct associations were not shown to be strong, especially for the female gender and low physical performance variables. Even so, it should be noted that in line with our results, previous studies, including systematic reviews found that female sex [9,14,16,17,18,21,35], a higher number of morbidities [9,21,36], and lower physical performance [11,37,38,39], were associated with fear of falling. Some studies have pointed out that these factors may be explained by a decrease in muscle mass, commonly seen in older age, the female sex, and individuals with morbidities. Fear of falling in women over 60 years of age has been associated with factors such as menopause, which may generate a decrease in bone mineral mass, a decrease in hormones, and consequent loss of muscle mass [40]. People with chronic diseases, such as diabetes and rheumatic diseases, can develop deficits in terms of mobility and pain, and a consequent restriction in the activities of daily life, which can lead to a decrease in muscle mass and fear of falling [41,42,43]. A previous systematic review and meta-analysis [44] found that most of the included studies showed that older adults of both sexes with fear of falling presented lower performance in the walking test, which can be explained by changes in gait phases, such as a shorter step distance, increased step width, and reduced gait speed.

The findings of the modeling in the present study indicate that older age, mediated by worse physical performance, is indirectly associated with greater fear of falling. It is worth noting that although the strength of the association is weak, this finding corroborates studies available in the literature, such as the study by Oh et al. (2017) [45]. In their study, Oh et al. (2017) [45] analyzed the physical performance of people over 65 years of age and found that older adults with limitations in lower limb exercises demonstrated approximately eight times more fear of falling than those without such limitations. Together, these results suggest that the presence of fear of falling may cause caution and avoidance of the activities of daily living and exercises in community-dwelling older adults; this can contribute to a reduction in muscle strength and worse physical performance, further aggravating the fear of falling, which may, therefore, establish a vicious cycle between increased risk of falling and reduced physical performance [46,47,48].

In this study, the number of falls was not significantly associated with the fear of falling. This contradicts the results of previous studies conducted in community-dwelling older adults [49,50]. A possible reason for this result is that there were few self-reports on the number of falls among the participants, which may be due to the participants’ lack of knowledge about the real definition of a fall (not only when reaching the ground). In a complementary way, it is highlighted that fear of falling can be present independently, that is, it can also exist in the absence of falling [11]. Another fact to be discussed is that fall-related self-efficacy has a high prevalence among community-dwelling older adults, and falling may be a consequence of older adults’ overestimated or underestimated perspective on their own physical condition, which may interfere with how they report or understand their fear of falling [4,51].

Psychological aspects such as fall-related self-efficacy, self-perception of health, and depressive symptoms or depression have been suggested to be associated with fear of falling [10,19,25,35,52,53]. Although the psychological processes that could explain fear of falling in older adults are still under investigation, when analyzing emotional regulation and fear of falling, Scarlett et al. (2018) [54] found depression to be a strong predictor of fear of falling in older adults. 

Similarly, older adults with poor self-perception of health tend to limit daily mobility to performing tasks and even stop practicing physical exercise, which may lead to poor physical performance and contribute to the appearance of fear of falling or its worsening [12,25,47]. Although we did not evaluate self-efficacy, overall, our results are in line with Bandura’s social learning theory related to fall-related self-efficacy, where performance mechanisms show a strong relationship with psychological changes and, consequently, behavioral modifications; this suggests that self-efficacy acts on the behavioral decisions of dispositions [10]. Therefore, similarly to physical factors, these psychological aspects could also have a bidirectional relationship with fear of falling, since fear of falling may increase depressive symptoms, and depressive symptoms are commonly associated with social isolation, as well as with a decrease in physical activity and exercise in this population [54,55]. In line with these results, our structural equation modeling analysis showed that worse self-rated health, mediated by a higher number of depressive symptoms, was associated with fear of falling. 

Of note, these physical and psychological aspects may act together in the underlying mechanisms of fear of falling and are probably preventable and treatable conditions. A negative perception of health in older adults has been associated with a greater demand for healthcare and more chronic diseases and morbidities, as well as factors that may contribute to a decrease in both physical activity and muscle mass, and to fear of falling [12,25]. A sedentary lifestyle is associated with the onset of chronic diseases and cardiovascular diseases, and decreased physical performance, and is a risk factor for mortality [56]. Despite this, the substitution of 30 min/day of sedentary time with activities of moderate to high intensity is associated with a decrease in fear of falling [57]. Therefore, the World Health Organization Guidelines on Physical Activity and Sedentary Behavior emphasize the health risks of physical inactivity, which should be avoided at all ages [39].

The cross-sectional design of the present study makes it difficult to establish causal relationships between the studied factors and the fear of falling, and future longitudinal studies are still needed to clarify the relationships between these factors. Another point is that the use of self-reported measures for clinical and health conditions and questionnaires may not precisely estimate some of the information found; moreover, to minimize possible biases in obtaining data via self-reporting, older adults with suggestive cognitive decline on the MMSE were excluded. On the other hand, this research included a large sample never before analyzed in a micro-region in the northern region of Brazil, and used the Falls Efficacy Scale International—Brazil (FES-I Brazil), a validated, reliable, and widely used instrument for assessing fear of falling. This is, to our knowledge, the first Brazilian study to use a structural modeling analysis approach to analyze factors associated with fear of falling in community-dwelling older adults, and our findings may help decision makers to develop strategies for managing fear of falling and preventing negative health outcomes in this population.

Thus, the data allow us to conclude that female sex, worse physical performance, a higher number of self-reported morbidities, and a higher number of depressive symptoms were directly associated with greater fear of falling. Additionally, older age, mediated by worse physical performance, as well as worse self-rated health, mediated by a higher number of depressive symptoms were associated with greater fear of falling.

## Figures and Tables

**Figure 1 jcm-12-00545-f001:**
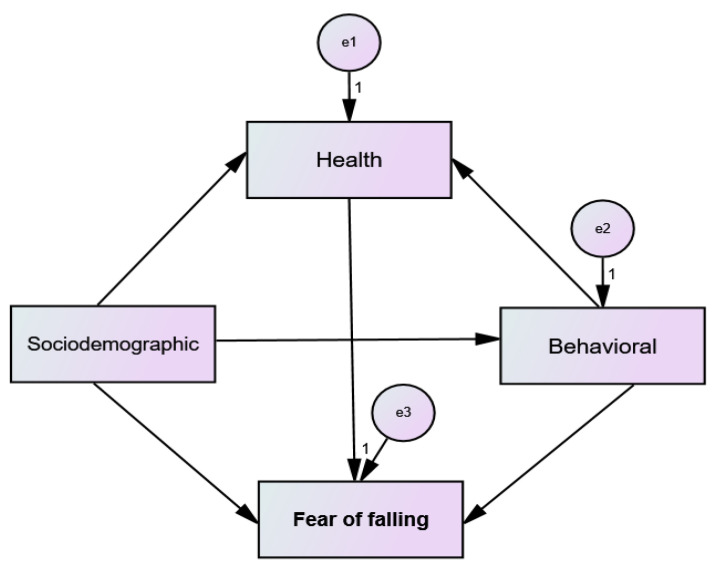
Hypothetical model tested.

**Figure 2 jcm-12-00545-f002:**
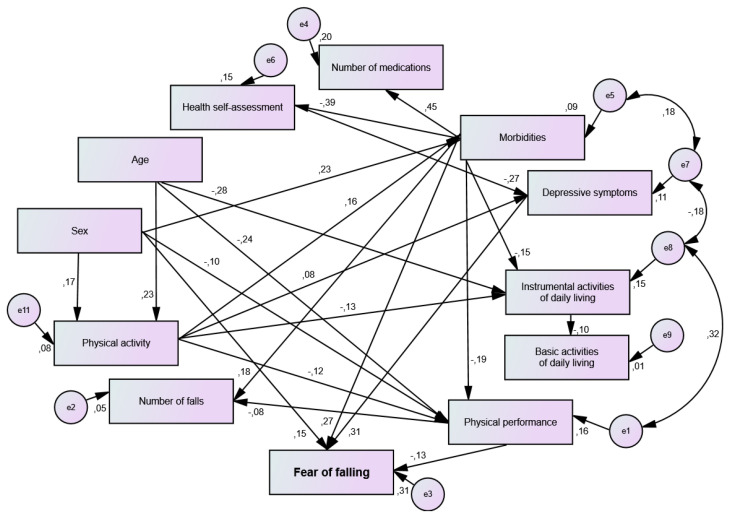
Model for analyzing the association between socioeconomic, clinical, and health characteristics and fear of falling among community-dwelling older adults (Macapá, AP, Brazil, 2017) (*n* = 410). (χ^2^ (gl = 41) = 54.78; *p* = 0.073; CFI = 0.95; GFI = 0.99; TLI = 0.93; RMSEA = 0.029).

**Table 1 jcm-12-00545-t001:** Characteristics of the community-dwelling older adults (Macapá, Amapá, 2017) (*n* = 410).

Variables	*n*	%
**Sex**		
Male	138	33.7
Female	272	66.3
**Health perception**		
Very bad	47	11.5
Bad	12	2.9
Regular	227	55.4
Good	95	23.2
Very good	29	7.1
**Physical activity level (IPAQ long version)**		
Sufficiently active (≥ 150 min/week)	192	46.8
Insufficiently active (≤ 149 min/week)	218	53.2
**Use of walking aid**		
Yes	17	4.1
No	393	95.9

IPAQ: International Physical Activity Questionnaire.

**Table 2 jcm-12-00545-t002:** Distribution of means and standard deviations of quantitative variables included in the structural model among community-dwelling older adults (Macapá, Amapá, Brazil, 2017) (*n* = 410).

Variables	Mean	Standard Deviation
Age (in years)	70.11	7.22
Number of morbidities	5.40	2.89
Number of medications	1.61	1.74
Number of falls	0.39	1.04
Depressive symptoms (GDS-15)	3.42	2.44
BADL (Katz Scale)	0.04	0.27
IADL (Lawton and Brody Scale)	18.32	2.81
Physical performance (SPPB)	9.24	1.96
Fear of falling (FES-I-Brazil)	23.59	7.16

BADL: Basic activities of daily living; GDS-15: Abbreviated Geriatric Depression Scale; SPPB: Short Physical Performance Battery; FES-I-Brazil: Falls Efficacy Scale International; IADL: instrumental activities of daily living.

**Table 3 jcm-12-00545-t003:** Direct standardized coefficients for variables associated with greater fear of falling among older adults in Macapá, Amapá, Brazil, 2017 (*n* = 410).

Direct Associations	Estimator	*p* *
Fear of falling		
Female sex	0.15	<0.001
Number of morbidities	0.27	<0.001
Depressive symptoms (GDS-15)	0.31	<0.001
Physical performance (SPPB)	0.13	0.003

** p* < 0.05.

## Data Availability

Data can be requested directly to the authors of this manuscript.

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
