# Peer review of "Investigating Factors Associated with Fear of Falling in Community-Dwelling Older Adults through Structural Equation Modeling Analysis: A Cross-Sectional Study"

_jcm, 2023, doi:10.3390/jcm12020545_

Round 1

Reviewer 1 Report

It was very interesting to focus on the fear of falling in community-dwelling older adults and the method of analysis. However, I was concerned about the following points, which led me to the present opinion.

Major points

Can 60 years old be considered older adults? According to the World Health Organization definition, it is 65 years of age or older. For this reason, the rationale for setting the age of the older adults at 60 needs to be clearly presented.

It is also questionable that, despite the fact that it is intended for people aged 60 and over, the cited references use content intended for people aged 65 and over.

Regarding age, the analysis was conducted on 410 subjects aged 70.11 ± 7.22. Is it correct to consider this age group to be the older adults as a whole, since there are differences in physical and cognitive functions between those in the first half of 60 and those in the second half of 70, and bias cannot be denied in the results obtained. From this point of view, it is necessary to clearly present the rationale behind the use of the age of 60 as the older adults age group.

 I appreciate the use of MMSE to confirm cognitive function in the assessment of cognitive function, but in the case of risk factors for falls, I believe that the presence of attention impairment as well as general cognitive dysfunction cannot be ignored.

Regarding the number of falls in the previous 12 months, you need to provide evidence that you are treating multiple falls and accidental falls together. Keywords says Accidental Falls, but isn't the higher risk of falls a case of multiple falls in a short period of time? I believe that the highest risk of falls is in cases where a person has had multiple falls in a short period of time. I also have the impression that the definition of Accidental Falls is unclear.

In community-dwelling older adults, self-efficacy may be too high, resulting in falls from movements that overestimate their own physical functioning, or they may underestimate their own physical functioning, resulting in falls from not maintaining the required amount of activity. This has been pointed out in many previous studies. In this study, it is necessary to present a view on this point.

There are too few references to study limitation; the content on cross-sectional design alone is insufficient.

Minor points

Words such as older adults and older people are mixed up. They need to be unified.

Author Response

Comments and Suggestions for Authors 1

It was very interesting to focus on the fear of falling in community-dwelling older adults and the method of analysis. However, I was concerned about the following points, which led me to the present opinion.

Major points

Point 1: Can 60 years old be considered older adults? According to the World Health Organization definition, it is 65 years of age or older. For this reason, the rationale for setting the age of the older adults at 60 needs to be clearly presented. It is also questionable that, despite the fact that it is intended for people aged 60 and over, the cited references use content intended for people aged 65 and over. Regarding age, the analysis was conducted on 410 subjects aged 70.11 ± 7.22. Is it correct to consider this age group to be the older adults as a whole, since there are differences in physical and cognitive functions between those in the first half of 60 and those in the second half of 70, and bias cannot be denied in the results obtained. From this point of view, it is necessary to clearly present the rationale behind the use of the age of 60 as the older adults age group.

Response 1: We thank you for your careful reading of this manuscript. In Brazil, according to our legislation (the Statute of the Elderly and the National Policy of the Elderly), older adults are people aged 60 years or older. The same is true for underdeveloped or developing countries, as is the case of Brazil.

Point 2:  I appreciate the use of MMSE to confirm cognitive function in the assessment of cognitive function, but in the case of risk factors for falls, I believe that the presence of attention impairment as well as general cognitive dysfunction cannot be ignored.

Regarding the number of falls in the previous 12 months, you need to provide evidence that you are treating multiple falls and accidental falls together. Keywords says Accidental Falls, but isn't the higher risk of falls a case of multiple falls in a short period of time? I believe that the highest risk of falls is in cases where a person has had multiple falls in a short period of time. I also have the impression that the definition of Accidental Falls is unclear.

Response 2: We thank you again for the important observations on these points in our manuscript. Indeed, we agree on the relationship between cognitive impairment and the occurrence of falls. Since our study investigated some conditions that were self-reported and depended on the older adult him/herself to respond, individuals with suggestive cognitive impairment assessed by MMSE were excluded from this sample. We have inserted this point as a limitation in the corresponding topic. Because of the analysis used in our study, we had to consider the number of falls from the initial question about the occurrence of falls in the last 12 months. Despite that, we agree with the questioning and appreciate this important observation.

Point 3: In community-dwelling older adults, self-efficacy may be too high, resulting in falls from movements that overestimate their own physical functioning, or they may underestimate their own physical functioning, resulting in falls from not maintaining the required amount of activity. This has been pointed out in many previous studies. In this study, it is necessary to present a view on this point.

Response 3: We are grateful for the comments. We added this point in the discussion section, lines 240-245.

Point 4: There are too few references to study limitation; the content on cross-sectional design alone is insufficient.

Response 4: The authors thank for the comments. More information has been added about study limitations, lines 280 and 284.

Minor points

Point 5: Words such as older adults and older people are mixed up. They need to be unified.

Response 5: We proceeded with the necessary changes (lines 23, 219 and 227).

Reviewer 2 Report

The authors have performed a study looking into the associations between fear of falling and several factors. The manuscript is well written and sufficiently detailed in the methods and results section. The statistical methodology used (SEM), provides the opportunity to look into the complex associations which are underlying fear of falling. The discussion section, although extensive and well written, could be extended further. Most of my comments are therefore related to the discussion section.

Minor revisions

The aim of the study described in the manuscript is formulated very exploratory and no conclusion is provided at the end of the discussion section. The manuscript would benefit from either; a) a more formal research question that can be answered in the conclusion, or b) the notion that the work was exploratory in nature and therefor no conclusions are yet to be drawn.

Lines 183-188: The strength of the indirect associations is very weak (all betas < 0.1). The authors should address this throughout their interpretations in the discussion section. For instance it seems in contrast to the statement in lines 212-215 which seems to suggest a strong association.

Table 3: Similarly, the direct associations are not very strong (all betas < 0.32). Specifically the associations with sex and physical activity are limited. Was this expected? Or is it similar to other literature. The authors should address the magnitude of these associations in the discussion section.

Smaller minor revisions

Line 24: “Data were collected using a structured form”, It is unclear to me what a structured form is. Please consider adding whether this was self-reported or obtained through an assessor.

Line 33 and 34: What is meant by “…based on the proposed analysis.” The analysis in the manuscript has been performed, it is no longer proposed. Please consider removing this final part of the the sentence.

Line 47-50: The authors suggest that there may be a vicious circle between FoF and Falls. The references do not fully support this idea: Schoene et al 2019 suggest that FoF may be at least partially independent of falls. Tomita 2018 only describes associations based on FoF and fall history (as opposed to prospectively monitored falls, which are likely less influenced by FoF due to recall bias). Pena et al 2019 may provide the evidence for the authors statement but I was unable to read the full text in Portuguese. Friedman et al 2002 (limitation: assessed fall history instead of prospectively) and Weijer et al 2021 (limitation: assessed a relatively physically strong population)  may be good references to also include.

Line 66: What is meant with “little explored regions”?

Line 242-245: The authors seem to suggest causal relations (“directly contribute to”) between several factors and fear of falling. However, the references provided are both cross-sectional studies. I suggest to rephrase the sentence to steer away from inferences about causality.

Figure 2: Number of falls is not associated to Fear of Falling. This is a surprising finding given the frequency of reports in literature about an association being present. The authors should comment on this finding in the discussion section.

Author Response

Comments and Suggestions for Authors 2

The authors have performed a study looking into the associations between fear of falling and several factors. The manuscript is well written and sufficiently detailed in the methods and results section. The statistical methodology used (SEM), provides the opportunity to look into the complex associations which are underlying fear of falling. The discussion section, although extensive and well written, could be extended further. Most of my comments are therefore related to the discussion section.

Minor revisions

Point 1: The aim of the study described in the manuscript is formulated very exploratory and no conclusion is provided at the end of the discussion section. The manuscript would benefit from either; a) a more formal research question that can be answered in the conclusion, or b) the notion that the work was exploratory in nature and therefor no conclusions are yet to be drawn.

Point 2: Lines 183-188: The strength of the indirect associations is very weak (all betas < 0.1). The authors should address this throughout their interpretations in the discussion section. For instance it seems in contrast to the statement (lines 212-215) which seems to suggest a strong association.

Point 3: Table 3: Similarly, the direct associations are not very strong (all betas < 0.32). Specifically the associations with sex and physical activity are limited. Was this expected? Or is it similar to other literature. The authors should address the magnitude of these associations in the discussion section.

Responses 1, 2 and 3: The authors are grateful for the pertinent considerations made by the reviewer. We inserted a sentence at the end of the discussion section reinforcing the findings and concluding the findings of the study. Likewise, we highlight the strength of the associations in the discussion section, according to your important notes. Please, find the changes in the manuscript as following: point 1, lines 292 to 296, point 2, lines 224 to 226, point 3, lines 206 and 207.

Point 1: 292 a 296 ; point 2: 224 e 226; point 3: 206 e 207.

Smaller minor revisions

Point 4: Line 24: “Data were collected using a structured form”, It is unclear to me what a structured form is. Please consider adding whether this was self-reported or obtained through an assessor.

Response 4: The authors are grateful for the reviewer's observations and have made the necessary changes to lines 24 and 25.

Point 5: Line 33 and 34: What is meant by “…based on the proposed analysis.” The analysis in the manuscript has been performed, it is no longer proposed. Please consider removing this final part of the the sentence.

Response 5: The authors removed the information at the end of the sentence, as recommended, line 34.

Point 5: Line 47-50: The authors suggest that there may be a vicious circle between FoF and Falls. The references do not fully support this idea: Schoene et al 2019 suggest that FoF may be at least partially independent of falls. Tomita 2018 only describes associations based on FoF and fall history (as opposed to prospectively monitored falls, which are likely less influenced by FoF due to recall bias). Pena et al 2019 may provide the evidence for the authors statement but I was unable to read the full text in Portuguese. Friedman et al 2002 (limitation: assessed fall history instead of prospectively) and Weijer et al 2021 (limitation: assessed a relatively physically strong population) may be good references to also include.

Response 5: The authors are grateful for the reviewer's comments. The authors understand that the expression "vicious cycle" is in fact not described in the cited references, as such references were inserted to reinforce the final outcomes arising from the adverse event fall (social isolation, increased risk hospitalizations and death) mentioned at the end of the paragraph. In view of the reviewer's observation, the authors opted for modifying the wording of part of the paragraph (lines 48 and 49) in order to improve readibility and clarify the meaning of the sentence for the readers.

Point 6: Line 66: What is meant with “little explored regions”?

Response 6: It refers to regions of a community, city or country, where there is no information about the outcome. As an example, we cite that in Brazil, studies on the prevalence of falls are concentrated in the South and Southeast metropolitan regions. Therefore, data on the prevalence of falls in the brazilian national context may not express the reality of the country, considering the scarcity of studies in other Brazilian regions.

Point 7: Line 242-245: The authors seem to suggest causal relations (“directly contribute to”) between several factors and fear of falling. However, the references provided are both cross-sectional studies. I suggest to rephrase the sentence to steer away from inferences about causality.

Response 7: The authors proceeded with the necessary modifications (lines 270 and 271).

Point 8: Figure 2: Number of falls is not associated to Fear of Falling. This is a surprising finding given the frequency of reports in literature about an association being present. The authors should comment on this finding in the discussion section.

Response 8: The authors are grateful for the reviewer's relevant observations. Comments on the results obtained in the analysis regarding the number of falls and fear of falling were added (lines 235 to 242).  

Round 2

Reviewer 1 Report

Thank you for your careful revisions.

I have the impression that it is easier to read than last time.

I would like you to correct the following

Minor points

Words such as p. 1 L35 Elderly and p. 3 L113 older residents are mixed up. They need to be unified.

Author Response

Minor points

Words such as p. 1 L35 Elderly and p. 3 L113 older residents are mixed up. They need to be unified.

Response 1: We are grateful for the observation. The authors added the necessary information in the text (p. 3 L113).
